# Impact of the COVID-19 Pandemic on Staging Oncologic PET/CT Imaging and Patient Outcome in a Public Healthcare Context: Overview and Follow Up of the First Two Years of the Pandemic

**DOI:** 10.3390/cancers15225358

**Published:** 2023-11-10

**Authors:** Andres Kohan, Sumesh Menon, Vanessa Murad, Seyed Ali Mirshahvalad, Roshini Kulanthaivelu, Adam Farag, Claudia Ortega, Ur Metser, Patrick Veit-Haibach

**Affiliations:** Toronto Joint Department Medical Imaging, University Health Network, Sinai Health System, Women’s College Hospital, University Medical Imaging Toronto, Toronto, ON M5T 1W7, Canada

**Keywords:** COVID-19, PET/CT, cancer, staging

## Abstract

**Simple Summary:**

Our aim was to identify if the COVID-19 pandemic delayed PET/CT staging among oncology patients. In this retrospective cohort of 1572 patients who underwent PET/CT before and during the COVID-19 pandemic, we did not identify statistically significant differences in the timing of staging or overall survival between groups. We, therefore, surmise that COVID-19 and the public health response to it did not significantly impact the diagnoses and outcomes of oncologic patients using PET/CT at our institution.

**Abstract:**

To assess the impact of the COVID-19 pandemic on the diagnosis, staging and outcome of a selected population throughout the first two years of the pandemic, we evaluated oncology patients undergoing PET/CT at our institution. A retrospective population of lung cancer, melanoma, lymphoma and head and neck cancer patients staged using PET/CT during the first 6 months of the years 2019, 2020 and 2021 were included for analysis. The year in which the PET was performed was our exposure variable, and our two main outcomes were stage at the time of the PET/CT and overall survival (OS). A total of 1572 PET/CTs were performed for staging purposes during the first 6 months of 2019, 2020 and 2021. The median age was 66 (IQR 16), and 915 (58%) were males. The most prevalent staged cancer was lung cancer (643, 41%). The univariate analysis of staging at PET/CT and OS by year of PET/CT were not significantly different. The multivariate Cox regression of non-COVID-19 significantly different variables at univariate analysis and the year of PET/CT determined that lung cancer (HR 1.76 CI95 1.23–2.53, *p* < 0.05), stage III (HR 3.63 CI95 2.21–5.98, *p* < 0.05), stage IV (HR 11.06 CI95 7.04–17.36, *p* < 0.05) and age at diagnosis (HR 1.04 CI95 1.02–1.05, *p* < 0.05) had increased risks of death. We did not find significantly higher stages or reduced OS when assessing the year PET/CT was performed. Furthermore, OS was not significantly modified by the year patients were staged, even when controlled for non-COVID-19 significant variables (age, type of cancer, stage and gender).

## 1. Introduction

The COVID-19 pandemic has impacted the past, present and, probably, future health of the general population via the direct consequence of contracting the virus. However, public health measures, the fear of contracting the disease and difficulty accessing health institutions due to reduced capacity have also possibly contributed to patient morbidity and mortality. More specifically, shutdowns and restrictive measures all over the world have been associated with an increase in the no-show rate at imaging centers [1], a decreased volume of imaging [1,2,3,4,5] and an increased stage or tumor burden [6,7,8,9,10,11,12,13,14] at the time point of imaging. Related to this, Maringe et al. [15] have evaluated different scenarios in the UK where the shutdown of screening programs, the delay in routine diagnostic imaging and the prioritization of symptomatic cases will probably account for 3291–3621 additional deaths attributed to four cancers (breast, lung, colorectal and esophageal) in the next 5 years. Furthermore, the National Cancer Institute in the US has estimated an additional 10,000 deaths from delayed screening in breast and colorectal cancer for the next 10 years [16].

In this context, one of the most relevant examinations in oncologic workup is [^18^F]-FDG PET/CT. The use of radioisotopes to detect neoplastic tissue is the standard of care for many diseases such as lung cancer, lymphoma, head and neck tumors and melanoma, among others [17]. Despite its important role in oncological staging and follow-up, PET/CT suffered the same fate as other imaging methods during COVID-19. An international survey conducted by the International Atomic Energy Agency in June and October 2020 showed a worldwide decrease in the number of PETs by 65.6% in June and by 40.3% in October [5]. Furthermore, the decrease in PET/CT was similar when compared to lung and breast cancer screenings [2,3,13,18].

Although there are several reports of head and neck cancer patients and other oncologic diseases presenting delays in diagnosis, higher tumor burden and higher stages [6,7,10,11,12,14,18], few of them focused on imaging. Additionally, some reports suggest that for the purpose of not delaying treatment, some patients skipped the imaging workup. On the other hand, we only found a few works addressing the staging of patients with PET/CT during the pandemic [19,20,21,22], and even fewer that considered the second year of the pandemic where diagnostic delays might have been more apparent [20,21,22]. It is, however, largely unknown if this impact was transient, limiting itself to the first year of the pandemic, or continued through the second year of the pandemic.

Our objective was, therefore, to evaluate if patients undergoing staging PET/CT in the first and second years of the pandemic had higher initial staging than those in the pre-pandemic era and to evaluate the possible impact on their OS.

## 2. Materials and Methods

This retrospective study was approved by the local ethics board, and proof of informed consent was waived accordingly.

### 2.1. Study Population

All patients over the age of 18 who were imaged with [^18^F]-PET/CT as part of their initial oncologic staging/diagnosis for either lymphoma, head and neck cancer, lung cancer or melanoma were included. All patients underwent primary staging without prior therapy. These indications were chosen for evaluation since they represent the largest referral group in our practice.

### 2.2. PET/CT Studies

All clinical PET/CT studies were performed using FDG as the radiotracer and as part of the patients’ routine clinical workup. FDG injection was performed 60 min prior to imaging, per previously published local institutional guidelines [23], with study acquisition tailored to the disease being studied. The imaging reports issued at the time of primary scanning were reviewed. All PET/CT studies were carried out in a tertiary university-affiliated cancer center.

### 2.3. Data Acquisition

A retrospective study of lung cancer, melanoma, lymphoma and head and neck cancer patients staged using PET/CT during the first 6 months of the years 2019, 2020 and 2021 were included for analysis. We chose to limit analysis to the first 6 months of the year. The pandemic was declared on 11 March 2020, and vaccination started in December of 2020. Hence, the most uncertainty for patients and healthcare workers/institutions was close to these dates, and it is where the biggest impact of COVID-19 on oncologic patients was anticipated.

After compiling the list of patients that fulfilled inclusion criteria, a systematic search in the radiology information system (RIS) database was performed to retrieve gender, date of birth, address and alive status. Other variables (i.e., stage at diagnosis, treatments, initial visit to referring physician, etc.) were additionally compiled from the patient’s medical chart. When documenting disease stage, the medical notes were compared to imaging reports to identify and correct possible staging errors, if required. Staging was performed according to current clinical guidelines. All clinical data were stored and tabulated for analysis. Two physicians (AK and SM) with more than 5 years of medical practice performed all the data handling and retrieval.

### 2.4. Statistical Analysis

Demographic data, medical history, cancer diagnosis, biopsy results, month and year the PET/CT was performed, time from oncologic visit to PET and from PET to treatment, cancer treatment history, disease status and clinical outcomes (progression- and relapse-free survival, as well as overall survival) were retrospectively collected. Continuous variables are presented as mean and standard deviation or median and interquartile ranges, depending on their distribution. Categorical variables are presented as proportions and confidence intervals.

Endpoints were chosen to subrogate for different variables that could have been affected by COVID-19. Distance traveled would help identify limited access to health facilities as patients would be forced to go outside of their region to seek treatment if COVID-19 measures forced health institutions to limit their services. Days from diagnosis/oncologic visit to PET and from PET to treatment would be affected if limited access to screening or reduced availability of PET were present during COVID-19 due to restrictions to offer those services or if treatment facilities were to prioritize certain patients over others (i.e., only urgent cases were evaluated and treated). For example, at our institution, non-urgent surgical procedures were delayed due to the need for reduced staff and patient traffic and the heightened sanitation protocols of the operating room between patients. Furthermore, it was also performed to reduce patient exposure to COVID-19, as hospitals were the institutions where the infected patients were being treated. Finally, staging could be affected by a number of variables beyond PET (effectiveness of screening, transport availability, lockdowns, etc.). But, to effectively modify staging from one year to the other, a large enough event/scenario needed to happen to alter any such variables. In this context, COVID-19 was the only event large enough worldwide to explain any possible shift in those variables and to effectively change the staging distribution at the moment of diagnosis. As our objective was to identify the extent of COVID-19’s impact on staging, if present, beyond the first year of pandemic, the specific culprit variable behind that impact is beyond the scope of this work and a subject of research for future studies.

To determine whether staging at the moment of diagnosis was impacted by COVID-19 restrictions, patients were grouped into 3 categories: pre-COVID-19 (1 January 2019 to 30 June 2019), 2020 COVID-19 (1 January 2020 to 30 June 2020) and 2021 COVID-19 (1 January 2021 to 30 June 2021). We compared the initial staging distribution between patients imaged in a pre-pandemic era and those imaged after the COVID-19 outbreak using a chi-squared or Fisher exact test, depending on expected cell values. Comparison of days to PET from diagnosis and distance traveled for imaging in all patients between groups was performed using Kruskal–Wallis or ANOVA, depending on their distribution. Tukey honestly significant difference (HSD) or post hoc Dunn’s tests were used when required. A univariate analysis of age at PET, initial stage, gender, type of tumor and the date of diagnosis for overall survival was performed. A multivariate analysis for overall survival was performed using Cox regression. All data analyses and tests were performed in R, and a *p* < 0.05 was considered statistically significant.

## 3. Results

Overall, 1572 patients who were initially staged for our targeted diseases between 2019 and 2021 (Figure 1) were included. The baseline description of our sample and characteristics by years can be seen in Table 1.

The distance traveled by the patients pre-, during and after pandemic initiation did not show a normal distribution, and the Kruskal–Wallis showed no significant difference (*p* = 0.39) when comparing between timeframes (Figure 2).

The days from the first oncologic visit to PET also showed a non-parametric distribution with a median of 11, 18 and 13 days for 2019, 2020 and 2021, respectively. The Kruskal–Wallis H tests determined that there was a statistically significant difference between the years (*p* < 0.05). The post hoc Dunn’s test using a Bonferroni corrected alpha of 0.017 indicated that the mean ranks of the following pairs were significantly different: 2019–2020, 2019–2021 and 2020–2021. Meanwhile, median days from PET to treatment were 19, 17 and 21.5 days for 2019, 2020 and 2021, respectively, with the Kruskal–Wallis H test indicating that there is a non-significant difference (*p* = 0.06) in the dependent variable between the different groups.

Overall survival using log-rank test determined that the stage, type of cancer and gender showed statistically significant differences (Appendix A), while the year the PET was performed was not different, as evidenced in the Kaplan–Meier curves (Figure 3). Age at diagnosis was also assessed for OS using the Wilcoxon test due to its non-parametric behavior evidencing a statistically significant difference (*p* < 0.05) between groups, where patients who died had a higher age compared to those who survived.

Finally, multivariate Cox regression including all significant OS variables and the year the PET was performed determined that lung cancer (*p* < 0.05 HR 1.76 CI95 1.23–2.53), stage III (*p* < 0.05 HR 3.63 CI95 2.21–5.98), stage IV (*p* < 0.05 HR 11.06 CI95 7.04–17.36) and age at diagnosis (*p* < 0.05 HR 1.04 CI95 1.02–1.05) had a significantly increased risk for death. Hence, whether PET was performed prior to or during the pandemic had no relationship to OS.

## 4. Discussion

In our study, we set out to identify the impact of COVID-19 on the use of PET/CT and on the patients’ initial staging and ultimately found that staging distribution was not significantly altered post-COVID-19. It has to be noted that the PET/CT service at our institution during the COVID-19 pandemic was never restricted.

We found that COVID-19 restrictions imposed in our country and province altered several parameters in terms of access to PET. For example, the number of PET scans significantly increased in the second COVID-19 year (Table 1). Also, the number of days from the first oncological visit to the PET/CT exam was longer in the first COVID-19 year. Finally, the number of initial staging PET/CT vs. restaging/follow-ups differed before and after the pandemic.

Currently, there are conflicting results published in the literature in regard to the effect of COVID-19 on the number of staging PET/CTs [20,21,22], ranging from no impact to decreased numbers of patients being scanned [1,4,5]. The partly contradictory results in different regions of the world are reviewed by Carvalho et al. [24], where out of 135 articles, only 53% had indicators referring to adequate cancer staging where stable or lower stages at diagnosis were observed. These differences should not surprise anyone, as the response to the pandemic was uneven between countries. Our results are actually two-fold. During the first pandemic year, our numbers were mostly stable/slightly decreased. This, in reality, however, might represent a significant decrease in the expected number of patients, as there is usually an incremental increase in the volume of PET/CTs performed, estimated at approximately 15% per year. Thus, the statistically increased number of scans in the second pandemic year could represent a catch-up effect.

We also found that the type of PET/CT requested changed with a significant increase in the number of referrals for staging (Table 1). This could reflect a change in referral trends from the referring physician; however, it could also be the consequence of seeing patients who did not show up for their follow-up scans once therapy was initiated. While patients certainly appreciate the importance of adequate initial staging, once therapy commenced, patients may have avoided follow-up imaging based on a perceived increased risk of coming to the hospital.

When comparing our results to the available literature, only Minamimoto et al. [20] found a shift towards increased staging in colorectal cancer. Staging vs. other indications remained stable in esophageal and lung cancer patients, and they actually found a decrease in staging among lymphoma patients. However, in their study population (64%), unlike ours (51%), restaging was already by far the most prevalent indication before the pandemic [20]. Also, in line with our results, Cao et al. [25] found no difference in the overall staging of head and neck cancer patients being studied with PET/MRI during COVID-19.

One indicator that was closely studied during the pandemic was the delay in access to healthcare. In our sample, we identified a significant increase in days from the first diagnosis to PET, with its peak in 2020. While our findings are in line with those published in Carvalho et al. [24], where 89% of the evaluated indicators showed a delay in access to diagnostic procedures, that delay did not impact the stage of disease (Table 1) or OS (Figure 3) in our population. It is documented in the literature that there is significant stage migration (i.e., lung cancer) when there is a delay between the diagnostic PET/CT and the start of therapy longer than 4 weeks [26]. As our delay remained under 3 weeks (from oncologic visit to PET and from PET to treatment), it could explain the stability in the stage. Minamimoto et al. [20] also reported stable staging during 2020, although it had a shift to higher stages in 2021 as opposed to our findings.

The stability of OS in our sample population throughout 2020 and 2021 is relevant, as it remained not significant (HR 0.89 CI95 0.74–1.07) when controlled for other non-COVID-19 significant variables (type of cancer, stage, age and gender). These results should be read in the context of our country (Canada), as the impact of COVID-19 in oncologic staging and OS is likely correlated to how governments handled the pandemic. This is supported in part by the literature [27] while contradicting other published studies [6,9,10,11,12].

Besides the number of studies, we also were interested in analyzing how general COVID-19 restrictions had possibly impacted the population we service, and as a surrogate to restricted access to healthcare, we analyzed the distance traveled by our patients. In concordance with what was published by Solis et al. [11] in a sample of patients with head and neck cancer, we did not find a statistically significant difference between the distance traveled by patients in the different pandemic years. However, while the distance traveled was not significantly increased, we mapped out in Figure 2 the median and average values of this indicator using the different months and years of this study and saw that the average was notably higher in 4/6 months of 2019. We hypothesize that this finding might refer to a non-significant number of people changing the habit of flying across the province to seek medical attention, which would have been extremely difficult in 2020, to use resources closer to them.

Our study has certain limitations that should be considered when interpreting our results. First, we conducted a retrospective evaluation, which subjects our results to the biases usually encountered in this type of study, although we tried to control for it using multiple sources for critical information such as death or initial staging. Another limitation is the type of population we serve, which may reflect selection bias. Our institution is a highly specialized oncologic center, which means that rural populations or patients with oncological diseases that do not require highly specialized care may not necessarily be referred to our center. Finally, we understand our population is not representative of every other country hit with the pandemic, but we believe that sharing our experience allows others to reflect on how to address future pandemic-like situations.

## 5. Conclusions

We found that patients undergoing PET/CT in the first and second years of the pandemic showed differences in several evaluated parameters, mostly in the type of exam (staging vs. other), with an increase in the number of PET/CTs performed in the second pandemic year and an increase in time interval between clinic and staging PET/CT, although without significant impact on stage or OS.

We did not find significantly higher stages or impaired OS when PET/CT was performed before or during the COVID-19 pandemic. Furthermore, OS was not significantly modified by the year patients were staged, even when controlled for non-COVID-19 significant variables (age, type of cancer, stage and gender).

## Figures and Tables

**Figure 1 cancers-15-05358-f001:**
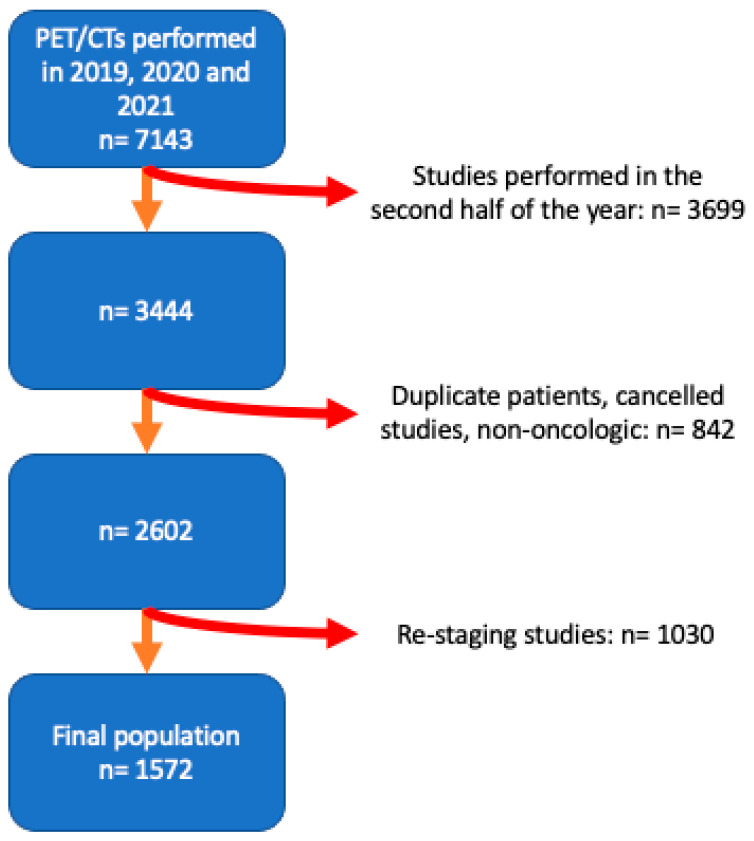
Patient flow from initial patient search to final study group.

**Figure 2 cancers-15-05358-f002:**
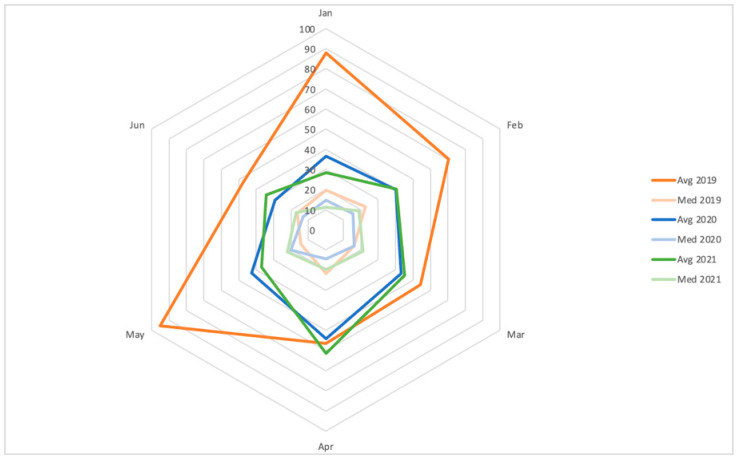
Distance traveled (km) expressed in mean (solid color) and median (faded color) during the first six months of 2019, 2020 and 2021.

**Figure 3 cancers-15-05358-f003:**
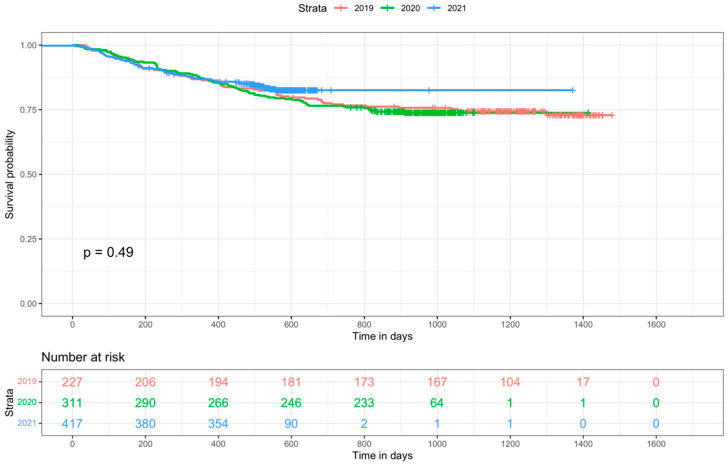
Kaplan–Meier curve comparing overall survival for patients staged with PET/CT in 2019 (orange), 2020 (green) and 2021 (blue).

**Table 1 cancers-15-05358-t001:** Population characteristics.

	2019–2021	2019	2020	2021	
**Overall Number of PET/CT**	2602	809	792	1001 *	*p* < 0.05
**Staging PET/CT n (%)**	1572	400 (49%) *	511 (65%) *	661 (66%) *	*p* < 0.05
**Gender n (%)**					
**Male**	915 (58%)	217 (54%)	316 (61%)	382 (58%)	*p* = 0.67
**Age median (IQR)**	66 (16)	65 (18)	65 (15)	68 (16)	*p* = 0.05
**Diagnosis n (%)**					
**Lymphoma**	512 (33%)	148 (37%)	155 (30%)	209 (32%)	*p* = 0.63
**Lung Cancer**	643 (41%)	160 (40%)	208 (41%)	275 (42%)	
**Melanoma**	81 (5%)	26 (6.5%)	26 (5%)	29 (4%)	
**Head and Neck**	336 (21%)	66 (16.5%	122 (24%)	148 (22%)	
**Staging n (%)**					
**I**		123 (31%)	168 (33%)	188 (28%)	*p* = 0.07
**II**		69 (17%)	88 (17%)	111 (17%)	
**III**		83 (21%)	105 (21%)	178 (27%)	
**IV**		121 (30%)	150 (29%)	181 (27%)	

* Significantly different when compared individually to the other years.

## Data Availability

Data are contained within the article and Appendix A.

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
