# Peer review of "Impact of the COVID-19 Pandemic on Staging Oncologic PET/CT Imaging and Patient Outcome in a Public Healthcare Context: Overview and Follow Up of the First Two Years of the Pandemic"

_cancers, 2023, doi:10.3390/cancers15225358_

Round 1
Reviewer 1 Report
Comments and Suggestions for Authors
This study is interesting and very well conducted, even from a statistical point of view.
I have no substantial comments.
But I have only a concern: in lines 172-173 you describe "the number of days from the first oncological visit to the PET/CT exam was no longer in the first COVID year (Table 1)", but I can't find this data in table 1. Can you explain? and eventually correct it?
Author Response
Thank you very much for your comments. You are right, it is not in the table. We initially had it in Table 1 but later decided to put it in the text (Lines 166-174). Hence, we removed the reference to Table 1.
Reviewer 2 Report
Comments and Suggestions for Authors
Manuscript title: "Impact of the COVID-19 pandemic on staging oncologic PET/CT imaging and patient outcome in a public healthcare context: overview and follow up of the first two years of the pandemic"
1) The study aimed to evaluate the effects of COVID-19 pandemic on cancer patient outcomes, such as distance traveled, PET/CT referral type, disease stage, overall survival.
2) The topic may be considered relevant by some readers.
3) The main study strength is detailed methodology, increasing the reproducibility. The main study weaknesses are possibly not clearly enough defined endpoints (i.e. rationale and data collection). For example, higher disease stage at PET/CT may be related to multiple factors including primary healthcare effectiveness (i.e., screening and early detection), transport availability, staff shortages. How was the data on distance traveled collected?
4) The conclusions are partially consistent with the goal provided, containing several additional outcomes not presented as study hypotheses.
5) The tables and figures are informative.
6) Specific comments are as follows:
- consider providing additional information behind the rationale for including each analyzed endpoint (i.e., miles traveled, days between staging and visit, scan type)
- consider providing a short sentence on primary study finding (i.e., if the null hypothesis was rejected) in the first paragraph of the Discussion
- consider informing the reader on the limitations of using "soft" study endpoints, which may be affected by multiple factors
Comments on the Quality of English LanguageConsider performing an additional language check using LanguageTool or similar application in order to further increase text readability.
Author Response
Thank you very much for your comments and suggestions. Please see below the extracted comments that required an answer and their respective answer immediately below
3) The main study strength is detailed methodology, increasing the reproducibility. The main study weaknesses are possibly not clearly enough defined endpoints (i.e. rationale and data collection). For example, higher disease stage at PET/CT may be related to multiple factors including primary healthcare effectiveness (i.e., screening and early detection), transport availability, staff shortages. How was the data on distance traveled collected?
The reviewer is correct in the sense that higher stages in PET/CT may relate to several factors. However, the endpoints analyzed were chosen to correlate with the variables the reviewer mentions and are known from the literature to have been impacted by COVID measures.
For example, distance traveled addresses the possibility of health facilities choosing/which needed to reduce health care services (PET imaging) due to pandemic measures. Thus, patients would have had to travel more frequently or farther to get imaged. Similar when evaluating day from diagnosis / study to treatment. Here, the potential delays in getting access to an oncologist, a surgeon or a radiation oncologist were clearly correlated with COVID restrictions as shown by several other publications.
Finally, staging, as you mention, could be affected by many variables, but for that endpoint to really shift, there needs to be a relevant enough scenario to shift significantly any of those variables and COVID was the only global event that transpired during that time that could logically explain such change.
A comment addressing this rationale has been added to the statistical analysis section in M&M (lines 119-136).
6) Specific comments are as follows:
- consider providing additional information behind the rationale for including each analyzed endpoint (i.e., miles traveled, days between staging and visit, scan type)
It has been included in Statistical analysis lines 119-136
- consider providing a short sentence on primary study finding (i.e., if the null hypothesis was rejected) in the first paragraph of the Discussion
We added the comment on the null hypothesis in the first paragraph of discussion.
- consider informing the reader on the limitations of using "soft" study endpoints, which may be affected by multiple factors
As mentioned above, we added more information about the "soft" endpoints in the M&M section. If required, such discussed points here can be integrated into the ‘Limitations” section of the study if the reviewer feels this is important for the reader’s understanding and overall quality of the manuscript.
- Consider performing an additional language check using LanguageTool or similar application in order to further increase text readability.
It has been read and corrected by a native English speaker.
Reviewer 3 Report
Comments and Suggestions for Authors
The manuscript goes further than previous reports to assess the impact of the COVID-19 pandemic on diagnosis staging and outcomes of oncologic disease from imaging, particularly FDG PET/CT perspective, The authors set out their objectives describe their methodology present their results and discuss them well.
In the abstract, the authors mentioned they used a convenience sample of patients who were staged for different cancers in the first six months of the years considered. I was trying to find out why it was limited to only the first half of the year.
It appears in the methodology however that they included patients who had FDG PET/CT throughout the whole year from 1st Jam to 31st Dec. Can the authors please explain this discrepancy? If they did only use the first half of the year then why?
Author Response
Thank you very much for your comments. You are correct, we missed the paragraph where the explanation to this particularity was given. We have added it to the methods section, and we also corrected the sentence that introduced confusion stating we included from Jan to Dec.
Round 2
Reviewer 2 Report
Comments and Suggestions for Authors
The authors have provided satisfactory responses to the reviewer's queries, further improving the manuscript.